# Kernel Function-Based Ambiguity Function and Its Application on DOA Estimation in Impulsive Noise

**DOI:** 10.3390/s22186996

**Published:** 2022-09-15

**Authors:** Yuzi Dou, Sen Li

**Affiliations:** College of Information Science and Technology, Dalian Maritime University, Dalian 116026, China

**Keywords:** *α*-stable distribution, DOA estimation, LFM signal, ambiguity function, time-frequency analysis

## Abstract

To solve the problem that the traditional ambiguity function cannot well reflect the time-frequency distribution characteristics of linear frequency modulated (LFM) signals due to the presence of impulsive noise, two robust ambiguity functions: correntropy-based ambiguity function (CRAF) and fractional lower order correntropy-based ambiguity function (FLOCRAF) are defined based on the feature that correntropy kernel function can effectively suppress impulsive noise. Then these two robust ambiguity functions are used to estimate the direction of arrival (DOA) of narrowband LFM signal under an impulsive noise environment. Instead of the covariance matrix used in the ESPRIT algorithm by the spatial CRAF matrix and FLOCRAF matrix, the CRAF-ESPRIT and FLOCRAF-ESPRIT algorithms are proposed. Computer simulation results show that compared with the algorithms only using ambiguity function and the algorithms only using the correntropy kernel function-based correlation, the proposed algorithms using ambiguity function based on correntropy kernel function have good performance in terms of probability of resolution and estimation accuracy under various circumstances. Especially, the performance of the FLOCRAF-ESPRIT algorithm is better than the CRAF-ESPRIT algorithm in the environment of low generalized signal-to-noise ratio and strong impulsive noise.

## 1. Introduction

Direction of arrival (DOA) estimation is a key field of array signal processing and has broad applications in radar, sonar, source localization, wireless communications, and other areas [1,2]. After decades of development, a relatively complete DOA estimation theory has been established. For stationary signal sources, the most frequently-used DOA estimation algorithms include the multiple signal classification (MUSIC) algorithm [3], and the estimation of signal parameters via rotational invariance techniques (ESPRIT) algorithm [4], which are based on the eigenvalue decomposition of the covariance matrix of the array received signal.

Many real-life signals, such as communication signals, radar, and sonar signals, have nonstationary characteristics. The linear frequency modulated (LFM) signal is a commonly used nonstationary signal that has ideal energy accumulation features in the time-frequency plane. In this context, the DOA estimation algorithms based on the covariance matrix suffer from performance degradation due to the inability to express the time-frequency domain characteristics of the LFM signal [5,6,7]. Time-frequency (TF) analysis is an important method to process nonstationary signals, so people try to combine the time-frequency analysis method into the DOA estimation algorithm. Professor Belouchrani first observed that the spatial Wigner-Ville distribution (WVD) matrix of the array received signals has a structure similar to the covariance matrix and applied it to estimate the DOA, thus TF-MUSIC algorithm was proposed [8]. Then, the TF-ESPRIT algorithm with lower computational complexity was proposed in [9,10] and the TF-MI-ESPRIT algorithm was built by using the multi-invariance (MI) characteristics to further improve the estimation performance [11]. In order to solve the problem that time-frequency points can’t be selected accurately in the low signal-to-noise ratio (SNR) environment, the array of received signals were firstly transformed to beamspace, and then DOA estimation was achieved by using the spatial WVD matrix of the beamspace signals in [12]. In order to improve the robustness of the spatial WVD matrix, spatial averaging and directional smoothing were used in [13]. Ambiguity function (AF) is another important quadratic time-frequency distribution [14], AF-MUSIC algorithm was proposed to estimate the DOA of narrowband LFM signals in [15]. Also based on AF, Ma and Goh studied the wideband DOA estimation problem in [16], and Xu et al. studied the MIMO radar target location problem in [17,18]. Other time-frequency analysis methods were also applied to DOA estimation, such as the short-time Fourier transform (STFT)-based DOA estimation algorithm was put forward in [19,20] and the fractional Fourier transform (FRFT)-based DOA estimation algorithm was proposed in [21].

The traditional DOA algorithms are mostly based on the assumption that the noise obeys the Gaussian distribution model. Under this assumption, analytically tractable solutions can be obtained by using the second-order statistics of the array received signals. However, the actual noise is often characterized by intensive impulsiveness with non-Gaussian distribution, such as sea clutter, forest clutter, atmospheric noise, and wireless channel noise [22]. Recent studies have found that these noises are more suitable to be described by *α*-stable distribution [23,24]. This basic achievement has been recognized by scholars in various research fields. Unfortunately, the *α*-stable distribution with characteristic exponent 0<α ≤ 2 does not have finite second-order statistics, thus the performance of the DOA estimation algorithms based on the covariance matrix is significantly reduced under the *α*-stable distribution noise environment.

To improve the performance of the covariance-based signal processing algorithm, the fractional lower order covariance (FLOC) was defined, and the FLOC-based subspace DOA estimation algorithm was proposed in [25]. Although the FLOC-based algorithms can effectively reduce the impact of impulsive noise, there are some limitations, such as the selection of fractional lower order parameter was highly dependent on the characteristic exponent of *α*-stable distribution which is difficult to obtain in practical application, and the FLOC-based algorithms are not suitable for strong impulsive noise environment. Due to the use of kernel function, correntropy became an effective method to compress impulsive noise [26]. A correntropy-based correlation (CRCO) was defined and a DOA estimation algorithm based on the CRCO matrix was proposed in [27]. By using the idea of the Hampel identifier, the authors of [28] expanded the correntropy to generalized correntropy (GCO) and used a GCO-based correlation matrix to estimate the DOA of coherently distributed sources under an impulsive noise environment. The kernel function used in correntropy is defined based on the second moment of the error signal, in order to further reduce the influence of the impulsive signal, the fractional lower order correntropy was defined by using the fractional lower order moment of the error signal in the kernel function [29,30]. In [31], a fractional lower order correntropy-based correlation (FLOCRCO) was defined and used to solve the DOA estimation problem in impulsive noise. Another important method to suppress impulsive noise is nonlinear preprocessing. The DOA estimation problem based on the bounded nonlinear preprocessing covariance matrix was studied in [32,33]. By combining the correntropy kernel function and nonlinear preprocessing method, a generalized correlation (GC) was defined and applied to estimate DOA in impulsive noise in [33].

The time-frequency analysis method and impulsive noise suppression method can be combined to solve the LFM signal processing problem in impulsive noise. Firstly, the ambiguity function and the fractional lower order statistics are combined to propose the fractional lower order ambiguity function (FLOAF) for joint estimation of time delay and Doppler shift [34]. The FLOAF was also used for solving the problem of MIMO radar localization under impulsive noise in [35]. Several time-frequency representations based on FLOC were defined and used for machine-bearing fault diagnosis under an impulsive noise environment in [36]. Then, Li et al. proposed a series of robust time-frequency analysis methods by using the sigmoid nonlinear preprocessing function [37,38,39]. In [37], a sigmoid-FRFT transform was proposed and applied to joint estimate the time delay and Doppler shift of the LFM signal in impulsive noise. A sigmoid-FRFT transform spectrum was further defined and applied to estimate the parameters of the LFM signal in [38]. The MIMO radar localization problem of the LFM signal was analyzed in [39] by using the sigmoid transform-based wideband ambiguity function.

However, currently, no studies have been reported about the combination of correntropy kernel function and the time-frequency analysis method. Therefore, in this paper, we firstly define the correntropy-based ambiguity function (CRAF) and fractional lower order correntropy-based ambiguity function (FLOCRAF) by combining the correntropy kernel function with the ambiguity function, and then their boundedness is analyzed. Next, the proposed two kernel function-based ambiguity functions are applied to solve the problem of DOA estimation of narrowband LFM signal sources in impulsive noise. By selecting multiple time-frequency points that are located in the energy accumulation region of the LFM signals to calculate the spatial CRAF matrix and FLOCRAF matrix, the DOAs are solved directly by applying the ESPRIT algorithm on the averaged CRAF matrix and FLOCRAF matrix, thus the CRAF-ESPRIT and FLOCRAF-ESPRIT algorithms are put forward. Computer simulation results show that the performance of the proposed algorithms is better than that of the algorithms only using the ambiguity function and the algorithms only using the correntropy kernel function-based correlation in terms of probability of resolution and estimation accuracy under various circumstances. For the two proposed algorithms, the FLOCRAF-ESPRIT algorithm is superior to the CRAF-ESPRIT algorithm in a low generalized signal-to-noise ratio and strong impulsive noise environment.

## 2. Preliminaries

### 2.1. *α*-stable Distribution

The probability density function (PDF) of *α*-stable distribution has a heavy tail, which provides a more effective theoretical tool for modeling the impulsive noise. As there is not a unified closed-form expression to describe it, the α-stable distribution is usually described by its characteristic function [23]:(1)$$\Phi \left(t\right)=\exp\left\{j\mu t-\gamma {\left|t\right|}^{\alpha }\left[1+j\beta \mathrm{sgn}\left(t\right)w\left(t,\alpha \right)\right]\right\}$$
where
(2)$$w\left(t,\alpha \right)=\{\begin{array}{c} \tan\left(\pi \alpha /2\right),\\ \left(2/\pi \right)\log\left|t\right|, \end{array}\begin{array}{c} \alpha \neq 1\\ \alpha =1 \end{array}$$
(3)$$\mathrm{sgn}\left(t\right)=\{\begin{array}{c} 1,\\ 0,\\ -1,\; \end{array}\begin{array}{c} t>0\\ t=0\\ t<0 \end{array}$$
where α (0<α≤2) is the characteristic exponent that controls the thickness of the tail of the PDF, smaller α leads to a heavier tail and hence the noise is more impulsive. β (−1≤β≤1) is symmetry parameter which is used to describe the slope of the distribution. When β=0, the α-stable distribution is symmetric about μ and is called symmetric α-stable (SαS) distribution. γ>0 is the dispersion parameter that behaves in a similar way to the variance of Gaussian distribution. −∞<μ<∞ is the location parameter. μ represents the median or mean when 0<α<1 or 1≤α≤2, respectively. In particular, the SαS distribution with α=2 is equivalent to the Gaussian distribution. For α-stable distribution with characteristic exponent α, its *p*th (p≥α) order moment does not exist.

The variance of α-stable distribution is infinite, thus the conventional signal-to-noise ratio (SNR) is no longer effective. Then the concept of generalized signal-to-noise ratio (GSNR) is defined as
(4)$$\mathrm{GSNR}={10\log}_{10}\left(\frac{\delta^{2}}{\gamma }\right)$$
where $\delta^{2}$ is the variance of signal.

### 2.2. Correntropy and Fractional Lower Order Correntropy

Based on the information theoretical learning and kernel methods, the team of Professor Principle extended the correlation function and proposed the concept of correntropy [26]. Correntropy can not only describe the similarity of two random variables but also suppress impulsive noise, which makes it more robust than the correlation function.

For any two random variables *X* and *Y*, correntropy is defined as:(5)$$V_{\sigma }\left(X,Y\right)=E\left[\kappa_{\sigma }\left(X-Y\right)\right]\;$$
where E[·] indicates the statistical expectation operator, $\kappa_{\sigma }\left(\cdot \right)=\frac{1}{\sqrt{2\pi }\sigma }\exp\left(-\frac{{\left|\cdot \right|}^{2}}{{2\sigma }^{2}}\right)$ is Gaussian kernel function based on the second-order statistics of the error signal X−Y, and σ is kernel width. In this context, a Taylor series of correntropy is expressed as: $V_{\sigma }\left(X,Y\right)=\frac{1}{\sqrt{2\pi }\sigma }\sum_{k=0}^{\infty }\frac{{\left(-1\right)}^{k}}{2^{k}{k!\sigma }^{2k}}E[{\left(X-Y\right)}^{2k}]$. In order to further suppress impulsive noise, fractional lower order correntropy is defined by combining kernel function with fractional lower order statistics of the error signal [29,30].

For any two random variables *X* and *Y*, fractional lower order correntropy is defined as:(6)$$V_{\sigma }^{p}\left(X,Y\right)=E\left[\kappa_{\sigma }^{p}\left(X-Y\right)\right]$$
where $\kappa_{\sigma }^{p}\left(\cdot \right)=\frac{1}{\sqrt{2\pi }\sigma }\exp\left(-\frac{{\left|\cdot \right|}^{p}}{{2\sigma }^{2}}\right)$. When the error signal X−Y is modeled by α-stable distribution, the parameter *p* is smaller than α. When p=2, the fractional lower order correntropy becomes the correntropy.

## 3. Proposed Robust Ambiguity Functions and Application on DOA Estimation

### 3.1. CRAF and FLOCRAF

Ambiguity function (AF) is one of the most popular time-frequency analysis methods, it was proposed by Professor Stein in [14]. The AF of nonstationary random signal s(t) is defined as
(7)$${\mathrm{AF}}_{ss}\left(\zeta ,\tau \right)=\int_{-\infty }^{+\infty }s\left(t\right)s^{*}\left(t-\tau \right)e^{-j2\pi \zeta t}dt$$
where ${\left(\cdot \right)}^{*}$ represents the conjugate operation, τ and ζ are time delay and frequency shift, respectively. AF can be regarded as the Fourier transform of instantaneous correlation function ${s(t)s}^{*}\left(t-\tau \right)$. When under the Gaussian noise environment, AF can reflect the signal energy distribution in the ambiguity domain. However, when in the case of SαS distribution noise environment for α<2, as the second-order correlation function is not bounded, it leads that AF cannot well reflect the signal energy distribution in the ambiguity domain. For this reason, the fractional low order ambiguity function (FLOAF) of nonstationary signal s(t) is defined as [34,35],
(8)$${\mathrm{FLOAF}}_{ss}\left(\zeta ,\tau \right)=\int_{-\infty }^{+\infty }s\left(t\right){\left[s^{*}\left(t-\tau \right)\right]}^{\langle p-1\rangle }e^{-j2\pi \zeta t}dt,\;p<\alpha$$
where ${\left[\cdot \right]}^{\langle p\rangle }={\left|\cdot \right|}^{\langle p-1\rangle }{\left(\cdot \right)}^{*}$. FLOAF can be regarded as the Fourier transform of instantaneous fractional low order correlation function $s\left(t\right){\left[s^{*}\left(t-\tau \right)\right]}^{\langle p-1\rangle }$. By adopting the fractional lower order statistics, FLOAF can reduce the influence of impulsive noise on the signal energy distribution in the ambiguity domain.

To enhance the robustness to impulsive noise, inspired by the correntropy kernel function-based correlation defined in [27,31], the definitions of two correntropy kernel function-based ambiguity functions are given below.

**Definition** **1.***Correntropy-based ambiguity function (CRAF) of nonstationary signal*s(t)*is defined as:*(9)$${\mathrm{CRAF}}_{\mathrm{ss}}\left(\zeta ,\tau \right)=\int_{-\infty }^{+\infty }\kappa_{\sigma }\left(s\left(t\right)-s^{*}\left(t-\tau \right)\right)s\left(t\right)s^{*}\left(t-\tau \right)e^{-j2\pi \zeta t}\mathrm{dt}$$*where CRAF is the Fourier transform of instantaneous CRCO function*$\kappa_{\sigma }\left(s\left(t\right)-s^{*}\left(t-\tau \right)\right)s\left(t\right)s^{*}\left(t-\tau \right)$. *CRAF can be regarded as the ambiguity function based on the Gaussian kernel function.*

**Definition** **2.**
*Fractional lower order correntropy-based ambiguity function (FLOCRAF) of nonstationary signal*

s(t)

*is defined as:*

(10)
$${\mathrm{FLOCRAF}}_{\mathrm{ss}}\left(\zeta ,\tau \right)=\int_{-\infty }^{+\infty }\kappa_{\sigma }^{p}\left(s\left(t\right)-s^{*}\left(t-\tau \right)\right)s\left(t\right)s^{*}\left(t-\tau \right)e^{-j2\pi \zeta t}\mathrm{dt},\;p<\alpha$$

*where FLOCRAF is the Fourier transform of instantaneous FLOCRCO function*

$\kappa_{\sigma }^{p}\left(s\left(t\right)-s^{*}\left(t-\tau \right)\right)s\left(t\right)s^{*}\left(t-\tau \right)$

*. When *

p=2

*, the FLOCRAF is the same as the CRAF. FLOCRAF can be regarded as the ambiguity function based on kernel function and fractional lower order statistics.*


**Property** **1.**
*CRAF and FLOCRAF are bounded:*

${\left|{\mathrm{CRAF}}_{\mathrm{ss}}\left(\zeta ,\tau \right)\right|}^{2}\left({\left|{\mathrm{FLOCRAF}}_{\mathrm{ss}}\left(\zeta ,\tau \right)\right|}^{2}\right)\leq \frac{E^{2}}{{2\pi \sigma }^{2}}$

*where*

$E=\int_{-\infty }^{+\infty }{\left|s\left(t\right)\right|}^{2}\mathrm{dt}$

*.*


The proof of Property 1 is given in Appendix A.

To verify the validity of the time-frequency focusing capability of the proposed ambiguity functions based on the correntropy kernel function, we consider a single linear frequency modulated (LFM) signal scenario. Let’s assume that the received signal s(t) is interfered by impulsive noise, it can be expressed as
(11)s(t)=r(t)+n(t) 
where ${r(t)=e}^{{j2\pi (f}_{0}t\;+\frac{1}{2}{\mathrm{vt}}^{2})}$ is LFM signal and the additive noise n(t) is modeled by the SαS distribution. $f_{0}$ and v are the initial frequency and chirp rate of the LFM signal, respectively. Figure 1 shows the AF, FLOAF, CRAF, and FLOCRAF of the signal s(t), given the GSNR=2 dB and the characteristic exponent of SαS distribution α=1.4. The detail parameters in r(t) are chosen as follows: $f_{0}=10\;\mathrm{Hz}$, v=20 Hz/s.

It is observed that AF fails to reflect the energy distribution of s(t) due to the influence of impulsive noise. However, in Figure 1b–d, the signal energy is mainly concentrated along the straight line ζ=20τ, which means that the FLOAF, CRAF, and FLOCRAF can reflect the energy distribution of s(t). In particular, CRAF and FLOCRAF have a better energy concentration performance than FLOAF, and FLOCRAF has the best energy concentration ability.

### 3.2. CRAF-ESPRIT and FLOCRAF-ESPRIT Algorithms

#### 3.2.1. Signal Model

Consider a uniform linear array (ULA) of *M* isotropic elements, and the distance between adjacent elements is d. Suppose that there are L (L<M) independent far-field narrowband LFM signals impinging on the array from different directions $\theta_{l},\;l=1,\;2,\;\ldots ,\;L$. Taking the first element as the reference point, the complex envelope of each signal received at the reference point is $s_{l}\left(t\right),\;l=1,\;2,\;\ldots ,\;L$, then the received signal at the *m*th element is defined as
(12)$$x_{m}\left(t\right)=\overset{L}{\underset{l=1}{\sum }}s_{l}\left(t\right)e^{-j\frac{2\pi }{\lambda }\left(m-1\right){d\sin\theta }_{l}}+n_{m}\left(t\right),\;m=1,2,\;\ldots ,\;M$$
where $n_{m}\left(t\right)$ is the independent identically distributed SαS distribution noise and is uncorrelated with the signals $s_{l}\left(t\right)$. The received signal x(t) can be expressed as vector,
(13)$$x(t)={\left[x_{1}\left(t\right),x_{2}\left(t\right),\;\ldots {,\;x}_{M}\left(t\right)\right]}^{T}=\mathrm{As}(t)+n(t)$$
where ${\left(\cdot \right)}^{T}$ represents the transpose operation. $s(t)={\left[s_{1}\left(t\right),s_{2}\left(t\right),\;\ldots {,\;s}_{L}\left(t\right)\right]}^{T}$ and $n(t)={\left[n_{1}\left(t\right),n_{2}\left(t\right),\;\ldots {,\;n}_{M}\left(t\right)\right]}^{T}$ are the source signals vector and noise vector, respectively. A is the M×L array steering matrix with $A_{\mathrm{ij}}={\;e}^{-j\frac{2\pi }{\lambda }\left(i-1\right){d\sin\theta }_{j}}$, and λ is the wavelength of the carrier.

#### 3.2.2. The Proposed DOA Estimation Algorithm

By using the impulsive noise suppression methods, scholars have proposed many DOA estimation algorithms under impulsive noise. These algorithms work well for stationary sources. However, these algorithms are not optimal for nonstationary signals with time-frequency characteristics, such as LFM signals. Making full use of time-frequency distribution characteristics of LFM signals can not only further improve the DOA estimation performance but also realize the DOA estimation with signal selectivity. In this section, we construct the CRAF matrix and FLOCRAF matrix to solve the DOA estimation problem of narrowband LFM signal under an impulsive noise environment.

Based on proposed CRAF and FLOCRAF, two M×M spatial correntropy kernel function-based ambiguity function matrices $R_{\mathrm{CRAF}}^{\zeta ,\tau }$ and $R_{\mathrm{FLOCRAF}}^{\zeta ,\tau }$ of array received vector x(t) can be obtained. The (*i*, *j*) entry of $R_{\mathrm{CRAF}}^{\zeta ,\tau }$ is defined as
(14)$$\begin{array}{cc} R_{\mathrm{CRAF}}^{\zeta ,\tau }\left(i,j\right) & =E\left[{\mathrm{CRAF}}_{x_{i}x_{j}}\left(\zeta ,\tau \right)\right]\\ & =E\left[\int_{-\infty }^{+\infty }\kappa_{\sigma }\left(x_{i}\left(t\right)-x_{j}{}^{*}\left(t-\tau \right)\right)x_{i}\left(t\right)x_{j}{}^{*}\left(t-\tau \right)e^{-j2\pi \zeta t}\mathrm{dt}\right] \end{array}$$

The (*i*, *j*) entry of $R_{\mathrm{FLOCRAF}}^{\zeta ,\tau }$ is defined as
(15)$$\begin{array}{cc} R_{\mathrm{FLOCRAF}}^{\zeta ,\tau }\left(i,j\right) & =E\left[{\mathrm{FLOCRAF}}_{x_{i}x_{j}}\left(\zeta ,\tau \right)\right]\\ & =E\left[\int_{-\infty }^{+\infty }\kappa_{\sigma }^{p}\left(x_{i}\left(t\right)-x_{j}{}^{*}\left(t-\tau \right)\right)x_{i}\left(t\right)x_{j}{}^{*}\left(t-\tau \right)e^{-j2\pi \zeta t}\mathrm{dt}\right] \end{array}$$

**Property** **2.**
*The CRAF matrix *

$R_{\mathrm{CRAF}}^{\zeta ,\tau }$

*and*
*FLOCRAF*
*matrix*

$R_{\mathrm{FLOCRAF}}^{\zeta ,\tau }$

*are uniformly represented by*

$R^{\zeta ,\tau }$

*, which can be expressed as*

$R^{\zeta ,\tau }={A\Lambda }^{\zeta ,\tau }A^{H}+\varepsilon^{\zeta ,\tau }I$

*, where*

$\Lambda^{\zeta ,\tau }$

*is a*

M×M

*diagonal matrix and*

$\varepsilon^{\zeta ,\tau }$

*is a constant.*


The proof of Property 2 is given in Appendix B.

Property 2 indicates that matrix $R^{\zeta ,\tau }$ can be decomposed into signal subspace and noise subspace, thus the matrix $R^{\zeta ,\tau }$ acts like the covariance matrix used in the subspace-based DOA estimation algorithm. The signal subspace can be obtained by eigenvalue decomposition of $R^{\zeta ,\tau }$ and then the DOA estimation can be realized by the ESPRIT algorithm. In this paper, the algorithms based on matrix $R_{\mathrm{CRAF}}^{\zeta ,\tau }$ and $R_{\mathrm{FLOCRAF}}^{\zeta ,\tau }$ are called CRAF-ESPRIT and FLOCRAF-ESPRIT, respectively.

In theory, $R^{\zeta ,\tau }$ in Property 2 is valid for each time-frequency point in the ambiguity domain. Since the LFM signal has the ideal energy accumulation features in the ambiguity domain, the time-frequency point in the energy accumulation region can be used to improve the accuracy of DOA estimation. In the scenario of multiple LFM signals with different chirp rates, the energy distributions of LFM signals are different in the ambiguity domain. Therefore, in order to estimate the DOAs of all LFM signals, it is necessary to select suitable time-frequency points for each LFM signal. When multiple time-frequency points of each signal are selected, multiple spatial time-frequency distribution matrices are obtained, it is necessary to average these matrices. Of course, DOA estimation with signal selectivity can also be realized by selecting the time-frequency points of a specific LFM signal.

The implementation steps are summarized as follows,

Step 1. Assume that the time-frequency characteristics of L independent LFM signals are known, and *K* time-frequency points are selected for each LFM signal. For every selected time-frequency point $\left(\zeta_{\mathrm{lk}}{,\tau }_{\mathrm{lk}}\right),\;l=1,\;2,\ldots ,\;L,\;k=1,\;2,\ldots ,\;K$, the spatial correntropy kernel function-based ambiguity function matrix $R^{\zeta_{\mathrm{lk}}{,\tau }_{\mathrm{lk}}}$ of array received signal x(t) is calculated.

Step 2. Average the spatial correntropy kernel function-based ambiguity function matrices, i.e., $R=\frac{1}{\mathrm{LK}}\sum_{l=1}^{L}\sum_{k=1}^{K}R^{\zeta_{\mathrm{lk}}{,\tau }_{\mathrm{lk}}}$.

Step 3. Perform eigenvalue decomposition on matrix $R^{\zeta_{\mathrm{lk}}{,\tau }_{\mathrm{lk}}}$ to obtain signal subspace.

Step 4. Apply the ESPRIT algorithm on the signal subspace to estimate DOA.

## 4. Simulation Results and Discussions

Consider a ULA with M=5 elements and the distance between adjacent elements is d=λ/2. Two narrowband signals impinge on the array from the directions $\theta_{1}=10^{{}^{\circ}}{\;\mathrm{and}\;\theta }_{2}=20^{{}^{\circ}}$. The waveforms of these two signals after down-conversion are LFM signals with $f_{01}=0.1\;\mathrm{Hz}$, $f_{02}=0\;\mathrm{Hz}$ and $v_{1}=20\;\mathrm{Hz}/s$, $v_{2}=-20\;\mathrm{Hz}/s$. The underlying noise is modeled as SαS distribution with characteristic exponent α and set the parameter p=α−0.1.

In this section, we firstly discuss the optimal selection of kernel width of the proposed CRAF-ESPRIT and FLOCRAF-ESPRIT algorithms, and then a series of numerical simulations are conducted to compare the performance of the proposed algorithms with the comparison algorithms, including ESPRIT, CRCO-ESPRIT, FLOCRCO-ESPRIT, GC-ESPRIT, AF-ESPRIT, and FLOAF-ESPRIT algorithms. Meanwhile, the simulations are performed with the change of four parameters, which are the GSNR, characteristic exponent α of the SαS distribution noise, the number of snapshots, and the angular separation of the two signals. Two performance indicators named root mean square error (RMSE) and probability of resolution are calculated for Q=500 Monte-Carlo experiments. The RMSE is defined as
(16)$$\mathrm{RMSE}=\frac{1}{2}\sum_{l=1}^{2}\sqrt{\frac{1}{Q}\sum_{q=1}^{Q}{\left({\hat{\theta }}_{\mathrm{lq}}-\theta_{l}\right)}^{2}}$$
where ${\hat{\theta }}_{\mathrm{lq}}$ is the *q*th estimated value of the real value $\theta_{l}$.

The probability of resolution $P_{a}$ is calculated by
(17)$$P_{a}=\left(\frac{1}{2Q}\sum_{l=1}^{2}Q_{\mathrm{sl}}\right)\times 100\%$$
where $Q_{\mathrm{sl}}$ is the number of successful experiments for the *l*th signal and an experiment is considered successful when it satisfies $|{\hat{\theta }}_{\mathrm{lq}}-\theta_{l}|\leq 1^{{}^{\circ}}$.

### 4.1. Simulations with Different Kernel Widths for the Proposed Algorithms

For the correntropy kernel function-based algorithms, the kernel width is an important factor affecting the performance. Given GSNR=5 dB and characteristic exponent α=1.2, 1.4, 1.6, 1.8, the influence of kernel width σ on the performances of CRAF-ESPRIT and FLOCRAF-ESPRIT algorithms are shown in Figure 2 and Figure 3, respectively.

It can be seen that in order to achieve a higher probability of resolution and lower RMSE, the optimal range of kernel width σ is (0.5, 1]. Thus, in the following simulation experiments, σ=0.6 is selected for the CRAF-ESPRIT algorithm, and σ=1 is selected for the FLOCRAF-ESPRIT algorithm.

### 4.2. Simulations with Different GSNRs

Considering the array received signals are interfered by SαS distribution noise with characteristic exponent α=1.6. When the GSNR ranges from −4 dB to 10 dB, the performances of ESPRIT, CRCO-ESPRIT, FLOCRCO-ESPRIT, GC-ESPRIT, AF-ESPRIT, FLOAF-ESPRIT, CRAF-ESPRIT, and FLOCRAF-ESPRIT algorithms are shown in Figure 4.

It can be seen that the RMSE decrease and the probability of resolution increase with the increase of GSNR for eight algorithms. Clearly, the performances of the AF-ESPRIT, FLOAF-ESPRIT, CRAF-ESPRIT, and FLOCRAF-ESPRIT algorithms which are utilizing the ambiguity function to extract time-frequency characteristics of signals are better than that of the algorithms without utilizing the time-frequency characteristics of signals, including ESPRIT, CRCO-ESPRIT, FLOCRCO-ESPRIT, and GC-ESPRIT algorithms. From the perspective of the impulsive noise suppression methods used in algorithms, the performances of CRAF-ESPRIT and FLOCRAF-ESPRIT (CRCO-ESPRIT, FLOCRCO-ESPRIT, and GC-ESPRIT) algorithms which use kernel function to inhibit impulsive noise are better than that of FLOAF-ESPRIT algorithm which use *p*th moment to suppress impulsive noise and AF-ESPRIT (ESPRIT) algorithm without utilizing impulsive noise suppression method. Among the algorithms utilizing the kernel function, the performance of the FLOCRAF-ESPRIT (FLOCRCO-ESPRIT) algorithm utilizing fractional lower order correntropy kernel function is superior to the CRAF-ESPRIT (CRCO-ESPRIT) algorithm utilizing correntropy kernel function. Therefore, the performance of the FLOCRAF-ESPRIT algorithm which is based on fractional lower order correntropy kernel function and time-frequency analysis method is the best, especially in a lower GSNR environment.

### 4.3. Simulations with Different Characteristic Exponents

To verify the robustness of the proposed algorithms to impulsive noise, the characteristic exponent of SαS distribution noise is set from strong impulsiveness α=1 to weak impulsiveness α=2, and GSNR=5 dB, the simulation results of eight algorithms are shown in Figure 5.

Clearly, the performances of the proposed CRAF-ESPRIT and FLOCRAF-ESPRIT algorithms are markedly better than the other six algorithms, especially in the strong impulsive noise environment. For example, when α=1.3, the probabilities of resolution of the two proposed algorithms can reach 100%, whereas the probabilities of resolution of the FLOAF-ESPRIT and AF-ESPRIT algorithms are only 44% and 3%, the probabilities of resolution of the ESPRIT, CRCO-ESPRIT, FLOCRCO-ESPRIT, and GC-ESPRIT algorithms are less than 1%. It is proved again that the correntropy kernel function-based ambiguity function can not only suppress impulsive noise but also extract the time-frequency characteristics of the signal. For the two proposed algorithms, the performance of the FLOCRAF-ESPRIT algorithm is better than the CRAF-ESPRIT algorithm.

### 4.4. Simulations with Different Number of Snapshots

In this experiment fixed α=1.6 and GSNR=5 dB, the simulation results in Figure 6 illustrate the performance of the ESPRIT, CRCO-ESPRIT, FLOCRCO-ESPRIT, GC-ESPRIT, AF-ESPRIT, FLOAF-ESPRIT, CRAF-ESPRIT, and FLOCRAF-ESPRIT algorithms under a wide range number of snapshots changes from 400 to 2000.

It can be seen that when the number of snapshots is bigger than 600, the probabilities of resolution of CRAF-ESPRIT and FLOCRAF-ESPRIT algorithms can reach 100%. However, even though when the number of snapshots is 2000, the probabilities of resolution of FLOAF-ESPRIT and AF-ESPRIT algorithms are only 98% and 86%, while the probabilities of resolution of the ESPRIT, CRCO-ESPRIT, FLOCRCO-ESPRIT, and GC-ESPRIT algorithms are less than 20%.

### 4.5. Simulations with Different Angular Separations

This experiment investigates the performances of the eight algorithms when $\theta_{1}$ is fixed to $10^{{}^{\circ}}$ and $\theta_{2}=10^{{}^{\circ}}+\Delta \theta$, where Δθ varies from $2^{{}^{\circ}}$ to $14^{{}^{\circ}}$ under α=1.6 and GSNR=5 dB. The simulation results of eight algorithms are shown in Figure 7.

It can be seen that the performances of all algorithms improve as the angular separation increases. This is because when the angular separation is relatively large, the signals are less cluttered with each other, which helps to estimate the DOA of the signal more accurately. For a certain angular separation, the performances of the two proposed algorithms are better than the other six algorithms, especially when the angular separation is relatively small.

## 5. Conclusions

Inspired by the effective impulsive noise suppression of correntropy kernel function, in this paper, two correntropy kernel function-based ambiguity functions, termed CRAF and FLOCRAF, are defined to well reflect the time-frequency distribution characteristics of LFM signal in an impulsive noise environment. Then, the two robust ambiguity functions are introduced to solve the problem of narrowband LFM signal source localization in an impulsive noise environment. Firstly, the spatial correntropy kernel function-based ambiguity function matrixes of the array received signal are calculated for the carefully selected time-frequency points, and then the ESPRIT algorithms are applied to obtain DOA estimation. Simulation results show that the performance of the two proposed algorithms is better than that of the algorithms utilizing only the ambiguity function to extract the time-frequency characteristic of signals but not using correntropy kernel function to suppress the impulsive noise and the algorithms only using correntropy kernel function to suppress the impulsive noise but not utilizing ambiguity function to extract the time-frequency characteristic of signals. For the two proposed algorithms, the performance of the algorithm based on fractional lower order correntropy kernel function is superior to the algorithm based on correntropy kernel function, especially in lower GSNR and stronger impulsive noise environment.

## Figures and Tables

**Figure 1 sensors-22-06996-f001:**
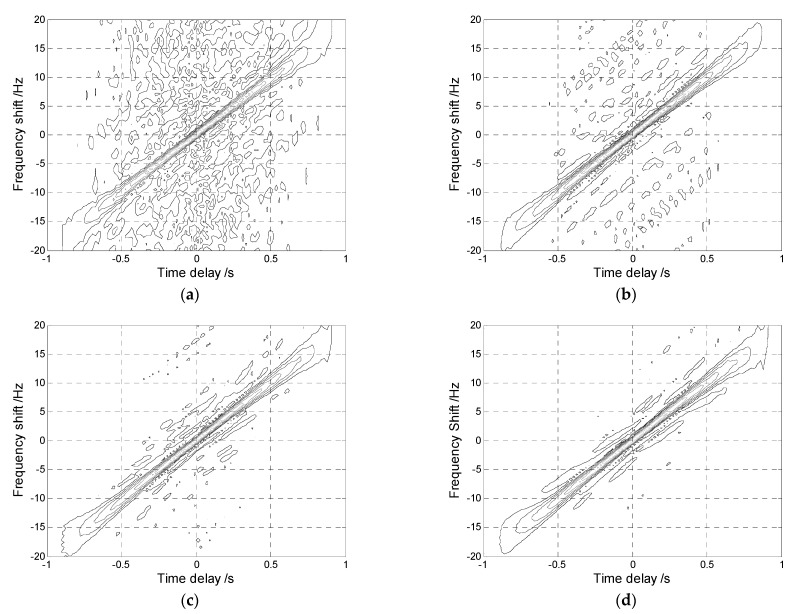
The time-frequency characteristic of LFM signal in SαS distribution noise with α=1.4: (**a**) AF; (**b**) FLOAF; (**c**) CRAF; (**d**) FLOCRAF.

**Figure 2 sensors-22-06996-f002:**
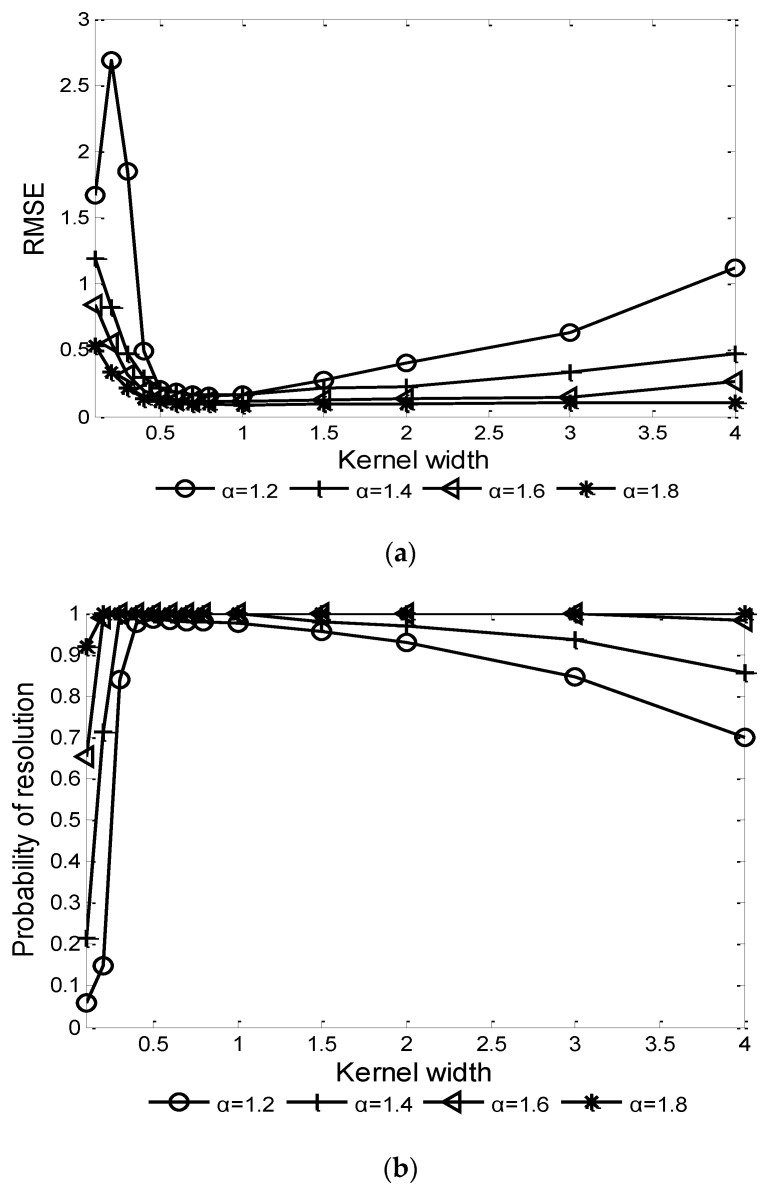
Performance of CRAF-ESPRIT algorithm vs. kernel width (GSNR=5 dB): (**a**) RMSE; (**b**) Probability of resolution.

**Figure 3 sensors-22-06996-f003:**
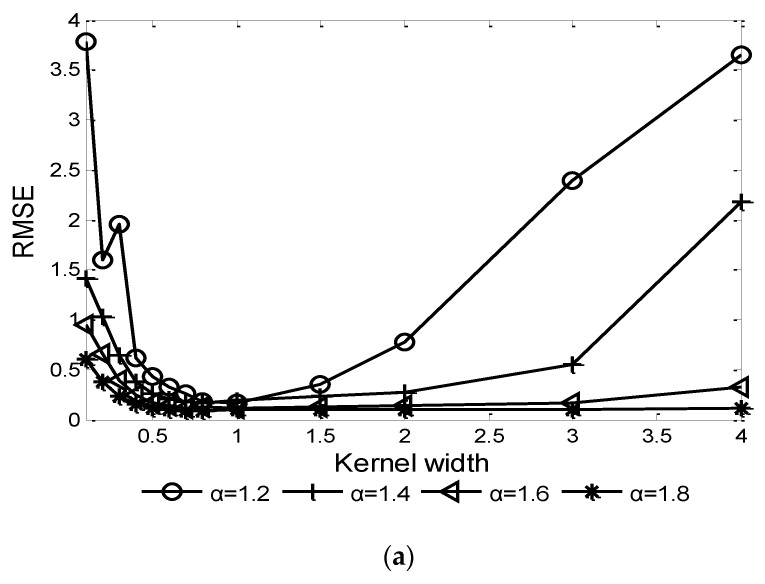
Performance of FLOCRAF-ESPRIT algorithm vs. kernel width (GSNR=5 dB): (**a**) RMSE; (**b**) Probability of resolution.

**Figure 4 sensors-22-06996-f004:**
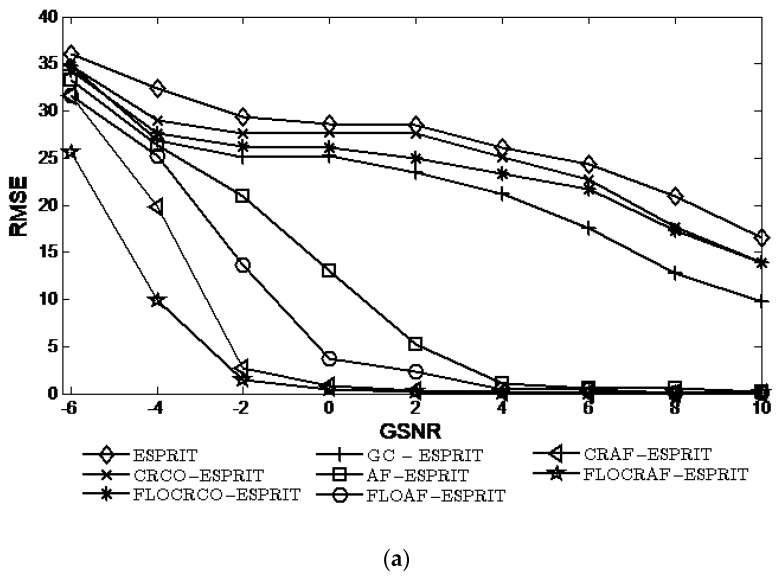
Performance vs. GSNR (α=1.6): (**a**) RMSE; (**b**) Probability of resolution.

**Figure 5 sensors-22-06996-f005:**
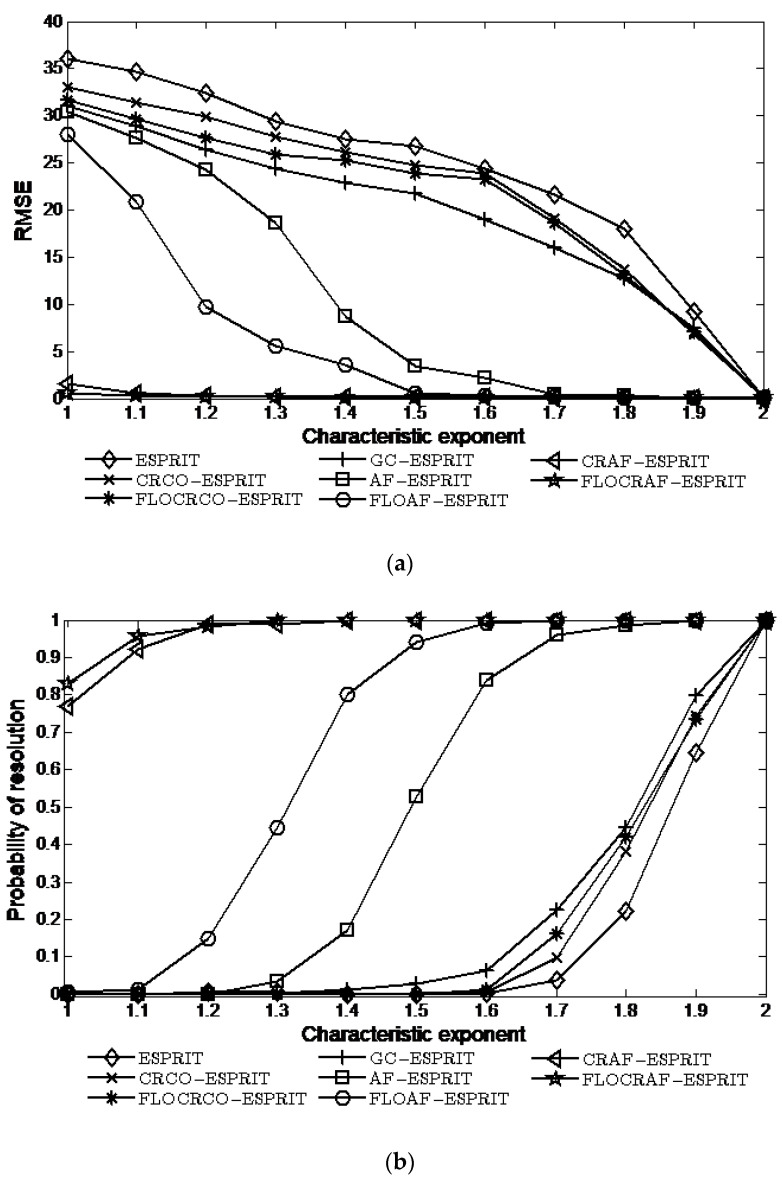
Performance vs. characteristic exponent (GSNR=5 dB): (**a**) RMSE; (**b**) Probability of resolution.

**Figure 6 sensors-22-06996-f006:**
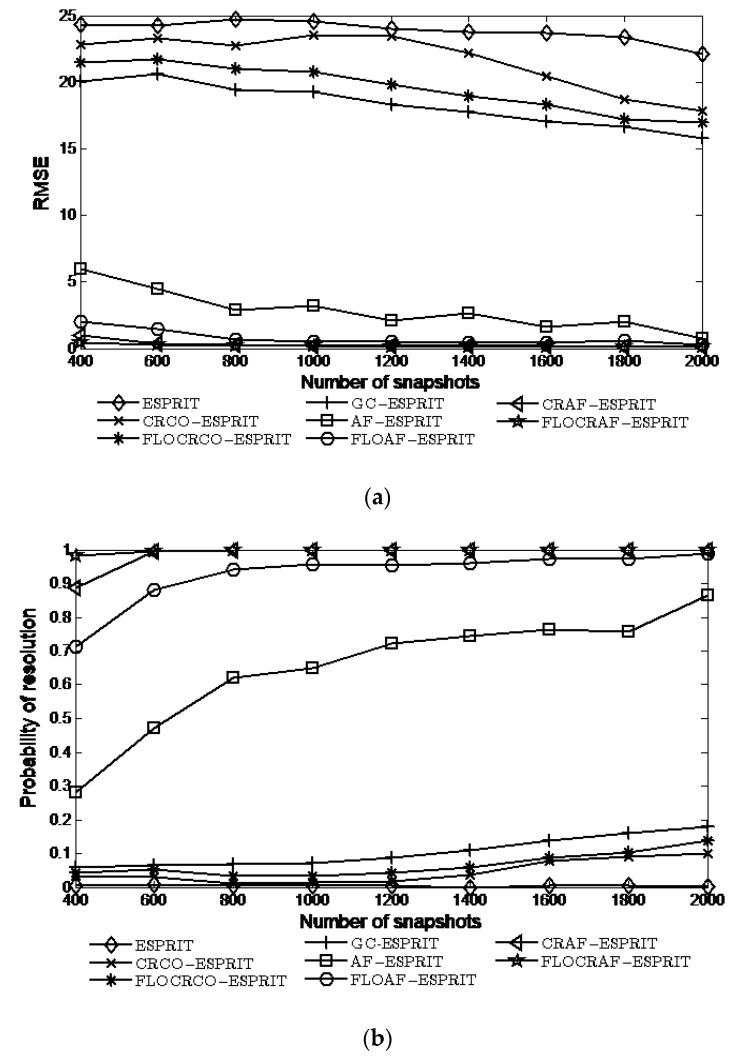
Performance vs. number of snapshots: (**a**) RMSE; (**b**) Probability of resolution.

**Figure 7 sensors-22-06996-f007:**
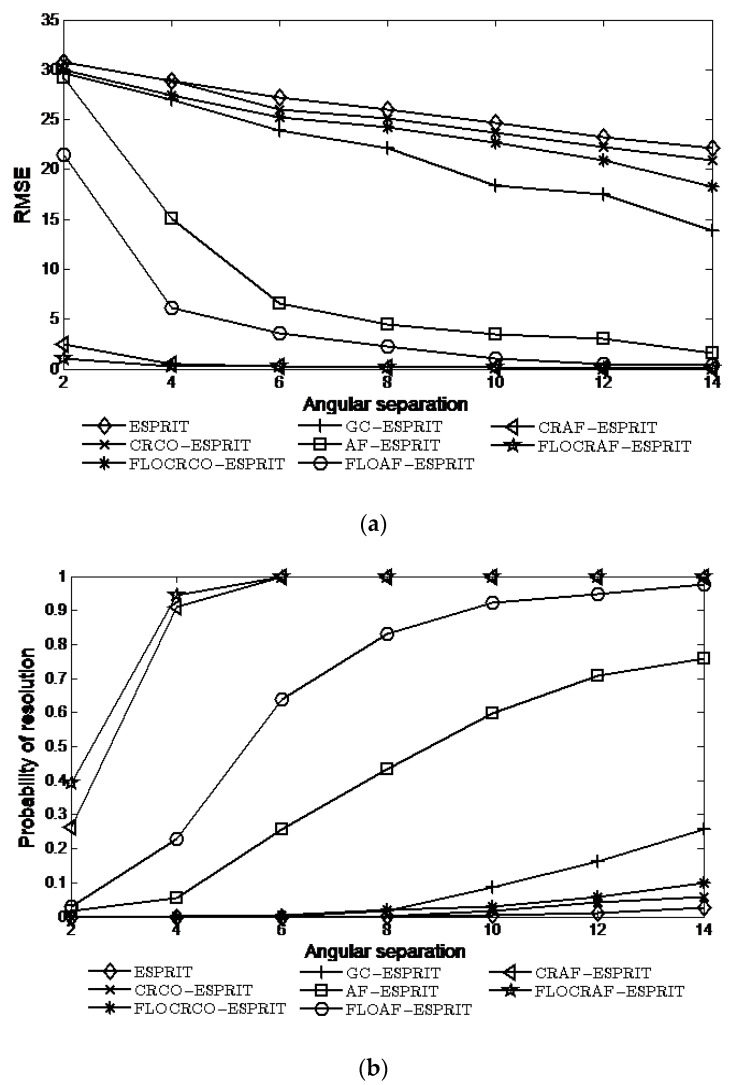
Performance vs. angular separation: (**a**) RMSE; (**b**) Probability of resolution.

## Data Availability

Not applicable.

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
