# Peer review of "Kernel Function-Based Ambiguity Function and Its Application on DOA Estimation in Impulsive Noise"

_sensors, 2022, doi:10.3390/s22186996_

Round 1

Reviewer 2 Report

The paper concerns the direction of arrival estimation (DOA) of narrowband signals with modulated frequencies in noisy conditions. The Authors propose the use of two methods for DOA estimation of narrowband LFM signals – CRAF-ESPRIT and FLOCRAF-ESPRIT. The applicability of the proposed approach is verified with the use of computer simulations. The paper is well organized and well written. The results are clearly presented, the plots correctly annotated, the conclusion well supported with the simulation results. However, some aspects of the paper still can be improved.

The paper would benefit from providing more details on the research process decisions, and addressing the following aspects:

-        The selection of the test signal (narrowband LFM signal) and noise parameters has not been convincingly justified. What is the justification for the selection of downconverted signal parameters in Section 4 Simulation Results and Discussions? (Is it possible to refer to any practical applications and signal examples?)

-        What was the criteria for selection of the number of uniform linear array elements in Section 4? What is the influence of the number of ULA elements on the performance of the proposed methods?

-        What is the justification for the selection of the range of angular separation of signals in Section 4?

-        What is the computational complexity of the proposed approaches compared to reference methods?

Reviewer 3 Report

1. Please try to explain the novel items of the paper. I think that although the combination of the two schemes has not yet been reported in the literature is indeed a motive, I am not so sure whether it can fulfill the scopes of the journal for the novelties.

2. Please try to be more specific and elaborate in the analysis of Section 3.

3. Finally, some more examples with even worse noise interactions would be welcome for the paper.

4. What are the limits of the proposed combination?

Round 2

Reviewer 3 Report

in this revised version the authors have managed to respond successfully to the majority of my comments. Therefore, I think that the manuscript can be accepted publication.